## [Peer Review File · Nature Genetics]

Allelic variation at a single locus distinguishes spring and winter faba beans

Corresponding Author: Dr Murukarthick Jayakodi

Version 0:

Decision Letter:

28th Jan 2025

Dear Dr Jayakodi,

Your Article, "Allelic variation at a single locus distinguishes spring and winter faba beans" has now been seen by 2 referees. You will see from their comments below that while they find your work of interest, some important points are raised. We are interested in the possibility of publishing your study in Nature Genetics, but would like to consider your response to these concerns in the form of a revised manuscript before we make a final decision on publication.

To guide the scope of the revisions, the editors discuss the referee reports in detail within the team with a view to identifying key priorities that should be addressed in revision. In this case, we think both referees have provided constructive reviews aimed at strengthening the analyses and improving the presentation, and we ask that you address their technical comments as thoroughly as possible with appropriate revisions. Please do not hesitate to get in touch if you would like to discuss these issues further.

We therefore invite you to revise your manuscript taking into account all reviewer and editor comments. Please highlight all changes in the manuscript text file. At this stage we will need you to upload a copy of the manuscript in MS Word .docx or similar editable format.

*2) If you have not done so already please begin to revise your manuscript so that it conforms to our Article format instructions, available

[here](http://www.nature.com/ng/authors/article_types/index.html).

*3) Include a revised version of any required Reporting Summary: <https://www.nature.com/documents/nr-reporting-summary.pdf>

EXTENDED DATA FIGURES

When re-submitting your manuscript, please ensure that any supplementary figures and tables that are crucial to the manuscript's conclusions are converted into Extended Data figures and tables to increase visibility of these data. Extended

Data figures and tables are online-only (present in the online PDF and full-text HTML versions of the paper), peer-reviewed display items that provide essential background to the article but are not included in the main article due to space constraints. A maximum of ten Extended Data display items (figures and tables) is permitted.

Link Redacted

We hope to receive your revised manuscript within 3 to 6 months. If you cannot send it within this time, please let us know.

Nature Genetics is committed to improving transparency in authorship. As part of our efforts in this direction, we are now requesting that all authors identified as 'corresponding author' on published papers create and link their Open Researcher and Contributor Identifier (ORCID) with their account on the Manuscript Tracking System (MTS), prior to acceptance. ORCID helps the scientific community achieve unambiguous attribution of all scholarly contributions. You can create and link your ORCID from the home page of the MTS by clicking on 'Modify my Springer Nature account'. For more information please visit please visit www.springernature.com/orcid.

Sincerely,
Wei

Wei Li, PhD
Senior Editor
Nature Genetics
www.nature.com/ng

Reviewers' Comments:

Reviewer #1 (Remarks to the Author):

Overall evaluation

=====

This manuscript presents a new improved version of the recently published faba bean reference genome Hedin/2. This manuscript also analyzes deeper the repetitive landscape as well as the centromeres. The manuscript strongest part is the analysis of more than 400 faba bean varieties focused on the frost tolerance trait. The manuscript also reuses some previously published population analysis to describe two GWAS that pointed to four different locations. The manuscript proposes some candidate genes associated to this trait although there is not functional validation. An extra support is based in an RNA-Seq experiment comparing two different varieties. Overall, it is an easy-to-read manuscript, with good selling points (new reference genome, centromere analysis, more than 400 accessions analyzed). Nevertheless, it lacks a strong functional validation due the limitations that this species poses. The authors did not use alternatives like pea which reduce the general impact of the manuscript. Still, the results reported in this manuscript not only represent a lot of work, it also could be relevant for the legume community working in frost tolerance.

Revised by Aureliano Bombarely on Dec. 1st, 2024.

Point-By-Point Manuscript Evaluation:

=====

A. Summary of the key results

The manuscript titled "Allelic variation at a single locus distinguishes spring and winter faba beans" describes a genetic study to identify candidate genes related with frost tolerance in faba beans. Additionally, it describes a new version of the Hedin/2 reference genome. The results are summarized in the following points:

- A new Hedin/2 reference genome was produced. It used optical mapping and HiC data to scaffold 97% of the assembly into 6 pseudomolecules. The number of gaps were reduced one order of magnitude (5,195 to 335). Only 295 Mb of the assembly remained unanchored. The QC (Mercury, BUSCO, LAI and duplications) of the new version (V2) was like the previous one (V1). Gene annotation was improved using Iso-Seq data, delivering 963 new gene models for a total of 35,107 gene models.
- A more detailed repetitive landscape analysis was produced. Most of the repetitive content for this faba bean genome is

produced by an expansion on the Ogre lineage of Ty/3 gypsy LTR-retroelements which accounts for the 71% of the genome. Centromere positions were identified by ChIP-Seq. Centromeres were big in size ranging from 6 to 20 Mb and composed mostly by FabTR.

- Genome-wide open chromatin regions were annotated. Different ATAC-Seq experiments were performed to annotate 121,443 and 93,501 accessible regions representing 39.6 Mb and 32.4 Mb in the adult leaf and seedling tissues, respectively.
- Genetic diversity analysis of spring and winter faba bean varieties. 209 and 197 faba bean spring and winter accessions were re-sequenced respectively (406 total) delivering almost 100M of SNPs. An admixture analysis revealed two distinctive clusters separating both types. Winter types showed a slightly higher nucleotide diversity ($\pi = 10.01 \times 10^{-3}$) than spring types ($\pi = 8.79 \times 10^{-3}$).
- Selective sweeps for improvement were identified. 47 selective sweep regions (11.03 Mb in total) were identified between both types. 228 genes were in those regions. Some of them, have GO terms associated with nodulation and cold acclimation pathways. The well-known VC1 locus was in one of these regions. New haplotypes of VC1 were identified in this analysis.
- Two different GWAS studies on 208 and 183 accessions. For the first study, two loci were identified on chromosomes 1 and 5 associated with winter hardiness. The first one overlapped with a peak previously described in a QTL analysis for frost tolerance. The GWAS analysis pointed the gene Vfaba.Hedin2.R2.1g002127 as significantly associated to this trait. A cold treatment showed that this gene and two other close genes annotated as CBF/DREB1 genes were induced as response to this stress. On chromosome 5, a chalcone synthase was identified as possible gene related with cold stress. For the second GWAS, two significant peaks were obtained for chromosomes 3 and 5, although the candidate genes were different from the first study (a homolog for FLK for chromosome 5 and small nuclear ribonucleoprotein for the chromosome 3).

B. Originality and significance: if not novel, please include reference

The manuscript is original, presenting a new improved faba bean Hedin/2 reference genome as well as some studies to identify genes associated to frost tolerance.

C. Data & methodology: validity of approach, quality of data, quality of presentation.

The approaches used in the manuscript looks correct as well as the quality of the data and the quality of the presentation.

D. Appropriate use of statistics and treatment of uncertainties

Yes, to my knowledge.

E. Conclusions: robustness, validity, reliability

Most of the conclusions are robust and valid with the data presented. Not sure if the description in the results of the first GWAS is right, because it mentions 180 spring type versus 28 winter type, in which case, it looks like an important unbalance (180/28) for the analysis. Additionally, there is not functional validation for the frost tolerance candidate gene relying on the GWAS, some published QTL results and the transcriptomic analysis of two varieties.

F. Suggested improvements: experiments, data for possible revision

The presented manuscript is a solid piece of scientific research. Nevertheless, some minor suggestions and recommendations are included in order to clarify/improve some parts:

- Genome completeness evaluation with BUSCO uses an obsolete version. The current BUSCO version is V5 as well as the embryophyta database is V10. Probably it will be adequate to use the specific fabales dataset v10 (fabales_odb10) too.
- Add other QC metrics for the genome annotation. The genome annotation could include other metrics such as % of genes with Swissprot/Trembl annotation as well as OMArk. Some comparative analysis with other fabid annotated genomes (e.g., using OrthoVenn3) could be useful. Finally, some extra screening of falsely annotated TE as genes could be useful (e.g., looking into TE related Interpro domains).
- Add more details for the Ogre lineage of Ty/3 gypsy LTR-retroelements. Being one of the most expanded elements in the faba bean genome, it would interesting to have more details about the evolutionary history of these elements if it was not treated in the previous version (if it was, please mention with a couple of lines referencing the previous version). Did these TE expand uniformly or do they have peaks of explosive growth? Are they located across the whole genome or are they focused on specific parts of the genome? How are the interactions between these elements and the gene models?
- More discussion about the origin of the gigantic faba bean chromosomes and centromeres could be interesting. The identification of the centromeres is an interesting results. Nevertheless, it looks to me that some parts about the chromosome evolution of this species have not been exploited. How the structure of the centromeres could influence the formation of these big chromosomes? Are unrelated features or the size of the centromeres is related with the size of the chromosomes?
- PCA structure does not look to support lower genetic diversity for winter types. In the figure 2A, the area of the winter type faba bean varieties is much more smaller than for the spring type. Even if the two main components accounts for less than 10% of the genetic variation, the PCA shows a dense cluster of winter type accessions. Additionally for $K > 2$, it can be appreciated some substructure of the spring types, in agreement what we can see in the PCA. This may not be incompatible with a higher nucleotide diversity, but I am wondering about possible reasons for this structure.
- Different number of VC1 copies undiscussed. The authors have mentioned that the Hedin/2 genome has 4 copies of VC1 meanwhile the Tiffany genome has 5. Do they know why? Are all the copies expressed?
- No functional analysis for the candidate genes. The functional validation of the candidate genes proposed in this

manuscript is based on a RNA-Seq experiment of two accessions and in some agreement with some QTL analysis. Knowing that the transformation of faba beans may be out of the scope of this manuscript, there are other close related systems such as pea that it could be used to have a better support on the candidate genes. Alternatively to extend the expression analysis by qRT-PCR on more accessions (at least 5 each type), could be an option thinking that the transformation times are quite long.

- Many details are missing in the material and methods. The material and methods have many missing details such as growth conditions for the plants used in each of the experiments (Optical mapping, CHIP-Seq, Iso-Seq...). The tissues used for the Iso-Seq and ATAC-seq were not specified either in this section. How the SPET data was obtained is not described (or the same citation than GBS should be added to SPET). There is not a good description of the panel of the different accessions used in this analysis. There is not description of how many replicates were used for the RNA-Seq experiments or how the libraries were prepared and the reads analyzed.
- Phenotype data is missing. The phenotype data should be included in the supplementary material.

G. References: appropriate credit to previous work?

Yes, to my knowledge.

H. Clarity and context: lucidity of abstract/summary, appropriateness of abstract, introduction and conclusions

The manuscript is clear and easy to read. The abstract as well as the introduction and the conclusions are adequate. The discussion needs a little bit more work in my opinion contextualizing these studies on other legumes.

Reviewer #2 (Remarks to the Author):

Zhang et al. improved a faba bean reference genome. Using resequencing and BSA data from two populations, identified a major winter hardiness locus and under selection. Overall, I think it is improved update over previous reference genome. But I have some major concerns or comments which the authors may consider to improve this research.

- 1) For the reference genome assembly, the continuity is improved via extra Bionano optical map data. The number of gaps decreased from 5,195 to 335, while the BUSCO is not significantly improved, indicating might the assembly is not significantly improved.
- 2) The authors highlighted the maximum intron size of 145 kb. As far as I know the maximum length of intron size is controversial. And the intron size results from settable parameters of splice-aware RNA-seq mapping program. Extra evidence or analysis are needed to confirm this large intron size.
- 3) Centromeres are enriched with repeat elements and difficult to assembly. To conduct comparison for centromeres size across chromosomes, the authors should investigate the assembly completeness of the centromere regions.
- 4) What is the relationship between the open chromatin regions with genome assembly or population analysis? What is the logic for putting the open chromatin analysis in the manuscript.
- 5) The author used a natural population with 406 accessions to conduct population analysis and highlighted this population as Fig. 2. Moreover, a selective sweeps analysis was performed and identified the VC1 locus is associated with a selection sweep during spring faba bean breeding. While to conduct GWAS analysis, the authors used a totally different population "ProFaba panel", without describing it population, where at least the LD decay is essential for GWAS analysis.
- 6) Natural selection is also conducted for the "ProFaba panel", where a Fst also identified two loci might under selection.
- 7) The d and e in Figure 3 (presumably) aim to indicate phenotypic differences between different types, but they lack statistical significance testing and annotations. The authors should review and add statistical tests for all similar instances in the manuscript.
- 8) In the analysis of cold tolerance, only 28 winter faba bean varieties were used, compared to 180 spring faba bean varieties. This may not fully represent the genetic diversity of winter varieties.
- 9) GWAS analysis identified DREB1 as the candidate genes, and the authors further checked the expression profile of this gene with low temperature treatment. This gene has been reported in diverse plant species, what is the novel discovery here?
- 10) It is unclear whether the two materials used for the frost transcriptome sampling are part of the population. Do these materials' genotypes correspond to the spring and winter haplotype classification within the population? Why were materials not selected directly from the population for transcriptome sequencing? Using only two materials' transcriptome data to represent the entire population is very one-sided.
- 11) The subsequent analysis of the cold tolerance GWAS results only speculates and describes a small number of candidate genes using transcriptome data, which is clearly insufficient. The authors should include experimental validation and additional relevant analyses to further clarify the impact of these genes on cold tolerance and winter survival.
- 12) Line 293, what is the data used for expression analysis? Please clarify.
- 13) Line 304, what is M1 and what is M2? Sound like the M2 model has more independent variables than M1, there is no wondering M2 improves prediction. Likelihood ratio testes could be performed to compare M1 and M2.
- 14) Line 307, the heritability and genetic structure are different for different traits. The strength of the correlations between prediction and observation for loss of leaf etc, could not tell the prediction performance for freezing tolerance.
- 15) Line 333, the syntenic conservation should be present in the result part.

Version 1:

Decision Letter:

19th Sep 2025

Dear Dr Jayakodi,

Your Article, "Allelic variation at a single locus distinguishes spring and winter faba beans" has now been seen by 2 referees. You will see from their comments below that while they find your work of interest, some important points are raised. We are interested in the possibility of publishing your study in Nature Genetics, but would like to consider your response to these concerns in the form of a revised manuscript before we make a final decision on publication.

We therefore invite you to revise your manuscript taking into account all reviewer and editor comments. Please highlight all changes in the manuscript text file. At this stage we will need you to upload a copy of the manuscript in MS Word .docx or similar editable format.

*2) If you have not done so already please begin to revise your manuscript so that it conforms to our Article format instructions, available

http://www.nature.com/ng/authors/article_types/index.html>here.

*3) Include a revised version of any required Reporting Summary: <https://www.nature.com/documents/nr-reporting-summary.pdf>

Please be aware of our <https://www.nature.com/nature-research/editorial-policies/image-integrity>>guidelines on digital image standards.

EXTENDED DATA FIGURES

Link Redacted

We hope to receive your revised manuscript within four to eight weeks. If you cannot send it within this time, please let us know.

Nature Genetics is committed to improving transparency in authorship. As part of our efforts in this direction, we are now requesting that all authors identified as 'corresponding author' on published papers create and link their Open Researcher and Contributor Identifier (ORCID) with their account on the Manuscript Tracking System (MTS), prior to acceptance. ORCID helps the scientific community achieve unambiguous attribution of all scholarly contributions. You can create and link your ORCID from the home page of the MTS by clicking on 'Modify my Springer Nature account'. For more information please visit please visit <http://www.springernature.com/orcid>>www.springernature.com/orcid.

Sincerely,
Wei

Wei Li, PhD
Senior Editor
Nature Genetics
www.nature.com/ng

Reviewers' Comments:

Reviewer #1 (Remarks to the Author):

Overall evaluation

=====

The revised version of manuscript has successfully resolved most of the concerns and questions risen during the first review. The authors have clarified the questions and have added deeper analysis about the QC for the genome annotations. Overall, it is an interesting manuscript, representing an incredible amount of work although a stronger functional validation of the candidate genes is missing. Nevertheless, due to faba beans are not a model species, and it has some important problems for validation, I believe that this manuscript has an important value (new genome assembly, proposal of candidate genes...) by its own.

Point-By-Point Manuscript Evaluation:

=====

A. Summary of the key results

See first revision.

B. Originality and significance: if not novel, please include reference

See first revision.

C. Data & methodology: validity of approach, quality of data, quality of presentation.

See first revision.

D. Appropriate use of statistics and treatment of uncertainties

See first revision.

E. Conclusions: robustness, validity, reliability

The authors clarified a little bit this part. Most of the conclusions are robust and valid with the data presented. Nevertheless, it is still not clear if the unbalance spring vs winter type (180/28) may be introducing some biases in the analysis analysis. The authors answered to the reviewer 2 "our panel includes over 10% (28 out 208) of winter-hardy genotypes, providing sufficient representation of winter growth habit related alleles. As a result, we were able to identify significant associations (Figure 3a)". Due the low diversity of the winter-hardy genotypes, it makes sense that it is going to produce a significant peak for the GWAS analysis (Figure 2a). Still the authors used a different population (Göttingen Winter Bean Population) to support their findings, but I found this part somewhat obscured in the manuscript. The qRT-PCR of other accessions is a good extra support, but again, somewhat is obscured and it does not appears clearly in the main manuscript.

F. Suggested improvements: experiments, data for possible revision

The authors have resolved all the points risen in this section, although an extra round of correction is recommended. There are some minor errors that should be fixed. For example line 485: "The BREAKER Long reads" should be BRAKER.

G. References: appropriate credit to previous work?

Yes, to my knowledge.

H. Clarity and context: lucidity of abstract/summary, appropriateness of abstract, introduction and conclusions

See the previous revision.

Revised by Aureliano Bombarely on Jun. 27th, 2025

Reviewer #2 (Remarks to the Author):

The authors had addressed my concerns and I have no further comments.

Version 2:

Decision Letter:

Our ref: NG-A67061R1

21st Nov 2025

Dear Dr. Jayakodi,

Thank you for submitting your revised manuscript "Allelic variation at a single locus distinguishes spring and winter faba beans" (NG-A67061R1). It has now been seen by the original referees and their comments are below. The reviewers find that the paper has improved in revision, and therefore we'll be happy in principle to publish it in Nature Genetics, pending minor revisions to satisfy the referees' final requests and to comply with our editorial and formatting guidelines.

Sincerely,

Wei Li, PhD
Senior Editor
Nature Genetics
New York, NY, USA
www.nature.com/ng

Reviewer #1 (Remarks to the Author):

The version NG-A67061R1 of the manuscript as well as the rebuttal letter has resolved and answered the concerns and questions risen in previous versions of this manuscript. There is not further aspect to review from my point of view and I endorse the publication of this manuscript.

Reviewers' Comments:

Reviewer #1 (Remarks to the Author):

Overall evaluation

=====

This manuscript presents a new improved version of the recently published faba bean reference genome Hedin/2. This manuscript also analyzes deeper the repetitive landscape as well as the centromeres. The manuscript strongest part is the analysis of more than 400 faba bean varieties focused on the frost tolerance trait. The manuscript also reuses some previously published population analysis to describe two GWAS that pointed to four different locations. The manuscript proposes some candidate genes associated to this trait although there is not functional validation. An extra support is based in an RNA-Seq experiment comparing two different varieties. Overall, it is an easy-to-read manuscript, with good selling points (new reference genome, centromere analysis, more than 400 accessions analyzed). Nevertheless, it lacks a strong functional validation due the limitations that this species poses. The authors did not use alternatives like pea which reduce the general impact of the manuscript. Still, the results reported in this manuscript not only represent a lot of work, it also could be relevant for the legume community working in frost tolerance.

Answer: We thank the reviewer for his positive feedback and for recognizing the practical limitations associated with carrying out functional validation in faba bean.

Point-By-Point Manuscript Evaluation:

The presented manuscript is a solid piece of scientific research. Nevertheless, some minor suggestions and recommendations are included in order to clarify/improve some parts:

- Genome completeness evaluation with BUSCO uses an obsolete version. The current BUSCO version is V5 as well as the embryophyta database is V10. Probably it will be adequate to use the specific fabales dataset v10 (fabales_odb10) too.

Answer: According to reviewer's suggestion, we rerun the BUSCO analysis with updated version using fabales_odb12 database and the newest BUSCO version 5.8.2 reported the results in Table 1 and supplementary table 5. The new results also emphasize the quality of our assembly results.

- Add other QC metrics for the genome annotation. The genome annotation could include other metrics such as % of genes with Swissprot/Trembl annotation as well as OMArk. Some comparative analysis with other fabid annotated genomes (e.g., using OrthoVenn3) could be useful. Finally, some extra screening of falsely annotated TE as genes could be useful (e.g., looking into TE related Interpro domains).

Answer: We appreciate reviewer's concern. We have now included the percentage of functional annotations from various databases (Nr, EggNOG, InterPro, Swissprot and trembl), along with OMArk annotation evaluation, in supplementary table 5, in comparison with v1. In addition, according to reviewer's suggestion, we ran OrthoVenn3 using seven legume species to identify homologous gene clusters and infer phylogenetic relationship, which are presented in supplementary figure 2. We also included the sentence (lines 140-142) to Results. TE-domain containing genes were removed in our annotation, but this was not previously mentioned in the method section. We have now clarified this omission.

- Add more details for the Ogre lineage of Ty/3 gypsy LTR-retroelements. Being one of the most expanded elements in the faba bean genome, it would interesting to have more details about the evolutionary history of these elements if it was not treated in the previous version (if it was, please mention with a couple of lines referencing the previous version). Did these TE expand uniformly or do they have peaks of explosive growth? Are they located across the whole genome or are they focused on specific parts of the genome? How are the interactions between these elements and the gene models?

Answer: Thanks for the question. We included the sentences (lines 154-170) to Results and prepared two new supplementary figures (suppl. fig. 3 and 4) to provide more information on this topic.

- More discussion about the origin of the gigantic faba bean chromosomes and centromeres could be interesting. The identification of the centromeres is an interesting results. Nevertheless, it looks to me that some parts about the chromosome evolution of this species have not been exploited. How the structure of the centromeres could influence

the formation of these big chromosomes? Are unrelated features or the size of the centromeres is related with the size of the chromosomes?

Answer: We agree that these are important and timely questions, however, they are hard to address using the available data. Our results from faba bean show that centromere size is not proportional to chromosome length. As described in the Results, line 185: "*Centromere size was not proportional to chromosome length, with the largest chromosome (chromosome 1) having the smallest centromere and the largest centromere being present on chromosome 6 which is the second smallest*". This observation rules out the hypothesis that the size of the centromeres scales with the chromosome length, as is particularly evident in chromosome 1, which is about twice as large as remaining chromosomes. Previous studies have reported significant variation in the morphology and sequence composition of centromeres in the genus *Vicia* and their close relative *Pisum* and *Lathyrus* species. Our results are consistent with these observations, as detailed in Results (lines 192-200). We believe that addressing the evolutionary and structural factors influencing centromere size and chromosome gigantism requires further comparative data. We anticipate that ongoing faba bean and *Pisum* pangenome projects will soon provide additional insights.

- PCA structure does not look to support lower genetic diversity for winter types. In the figure 2A, the area of the winter type faba bean varieties is much more smaller than for the spring type. Even if the two main components accounts for less than 10% of the genetic variation, the PCA shows a dense cluster of winter type accessions. Additionally for $K > 2$, it can be appreciated some substructure of the spring types, in agreement what we can see in the PCA. This may not be incompatible with a higher nucleotide diversity, but I am wondering about possible reasons for this structure.).

Answer: The winter diversity panel used in this study is a MAGIC-like (multi-parent advanced generation inter cross) population derived from 11 winter-type founder lines that were open pollinated for eight generation, commonly referred to as the "Göttingen Winter Bean Population (GWP)." This genetic structure is reflected in the PCA results, where the winter genotypes form a relatively tight cluster compared to the more diverse spring types. Notably, the open pollination of winter population resulted in higher diversity than spring lines, which exhibit relatively low outcrossing rate (Link et al. 1994). Whereas, the spring-type elite faba beans included in this study consist of elite breeding material drawn from the narrow European gene pool and from a well recombined population, so each of them shares a bit of genome with each of them. The visible population substructure among spring types corresponds to regional differentiation, with accessions originating from Northern, Central, and Southern Europe reflecting localized selection of alleles.

- Different number of VC1 copies undiscussed. The authors have mentioned that the Hedin/2 genome has 4 copies of VC1 meanwhile the Tiffany genome has 5. Do they know why? Are all the copies expressed?).

Answer: We appreciate the reviewer raising this point. In response, we have added Extended Figure 7e and Supplementary figure 12 to compare the expression patterns and sequence alignments of four VC genes in Hedin/2 with 5 genes in Tiffany. The data reveal that three VC1 paralogs are transcriptionally active in Hedin, whereas only two are expressed in Tiffany. This observation aligns with previous reports (Björnsdotter et al. 2022) demonstrating that AT insertions in the exonic regions of VC-1a and VC-1b induce frameshift mutations. Such structural alterations disrupt the encoded protein's reading frame, ultimately leading to loss of function in these alleles. There is also a tandem duplication deletion in the 3rd intron and SNPs for all VC genes in Tiffany, which may affect the function of the protein.

- No functional analysis for the candidate genes. The functional validation of the candidate genes proposed in this manuscript is based on a RNA-Seq experiment of two accessions and in some agreement with some QTL analysis. Knowing that the transformation of faba beans may be out of the scope of this manuscript, there are other close related systems such as pea that it could be used to have a better support on the candidate genes. Alternatively to extend the expression analysis by qRT-PCR on more accessions (at least 5 each type), could be an option thinking that the transformation times are quite long.

Answer: We thank reviewer for the understanding and for providing a practical suggestion. We performed qRT-PCR analysis for 14 CBF/DREB1 genes on chromosome 1 QTL region using five cold tolerant and susceptible faba accessions. The result is consistent with RNA-seq results and is now shown in Supplementary figure 13 with primer sequences are included in Supplementary table 24.

- Many details are missing in the material and methods. The material and methods have many missing details such

as growth conditions for the plants used in each of the experiments (Optical mapping, CHIP-Seq, Iso-Seq...). The tissues used for the Iso-Seq and ATAC-seq were not specified either in this section. How the SPET data was obtained is not described (or the same citation than GBS should be added to SPET). There is not a good description of the pannel of the different accessions used in this analysis. There is not description of how many replicates were used for the RNA-Seq experiments or how the libraries were prepared and the reads analyzed.

Answer: Thanks for pointing this out. We have included the details in line 512-516 (ATAC-seq), In-374 (Optical map), In-458 (Iso-Seq) and 645 (RNA-seq).

• Phenotype data is missing. The phenotype data should be included in the supplementary material.

Answer: Thanks for pointing this out. We added the phenotype data in the supplementary table 18 and 23.

Reviewer #2 (Remarks to the Author):

Zhang et al. improved a faba bean reference genome. Using resequencing and BSA data from two populations, identified a major winter hardiness locus and under selection. Overall, I think it is improved update over previous reference genome. But I have some major concerns or comments which the authors may consider to improve this research.

1) For the reference genome assembly, the continuity is improved via extra Bionano optical map data. The number of gaps decreased from 5,195 to 335, while the BUSCO is not significantly improved, indicating might the assembly is not significantly improved.

Answer: Retrotransposons in faba bean, such as OGRE, are over 20–30 kb in length (our first faba genome paper by Jayakodi et al., 2023), exceeding the average HiFi read length (16–18 kb). As a result, faba version 1 genome contained thousands of gaps. Instead of generating additional HiFi data, we employed bionano optical mapping, which significantly improved assembly contiguity, particularly in difficult-to-assemble regions rich in recent retrotransposons and megabase-scale satellite repeats. This is indeed a significant improvement. Moreover, BUSCO effectively assesses gene space and it does not verify efficiently in complex noncoding regions. Therefore, we also used complementary tool such as Merqury to validate the assembly quality. Additionally, we successfully assembled tandemly duplicated key genes (Extended Data Fig. 5a) and anchored over 400 Mb of previously unanchored sequence to chromosomes compared to version 1 (Supplementary Table 2). The contiguity also increased the RNA-seq read mappability and allowed us to annotate new genes in this assembly (supplementary Table 5). From genomics perspective, this is substantial improvement considering the genome size and complexity of faba bean.

2) The authors highlighted the maximum intron size of 145 kb. As far as I know the maximum length of intron size is controversial. And the intron size results from settable parameters of splice-aware RNA-seq mapping program. Extra evidence or analysis are needed to confirm this large intron size.

Answer: We agree with the reviewer that intron size is tricky particularly only with de novo prediction and RNA-seq data. However, we used full-length mRNA long read information to identify long introns. We added a new supplementary figure 13 showing the Iso-seq data support for long intron of 145 kb.

3) Centromeres are enriched with repeat elements and difficult to assembly. To conduct comparison for centromeres size across chromosomes, the authors should investigate the assembly completeness of the centromere regions.

Answer: To answer this question, we took advantage of the previous study by Avila Robledillo et al. (2018; <https://doi.org/10.1038/s41598-018-24196-3>), which analyzed the composition of satellite DNA in *Vicia faba* using clustering-based identification of repeat sequences from unassembled shotgun reads. Since this study was based on unassembled reads, it provides a valuable benchmark for evaluating the repeat completeness of our assembly. Seven families of centromeric satellite repeats were identified in this prior study, and all are present in our assembly. Additionally, our data reveal the centromeric localization of several other repeat elements (Supplementary Table 11). Furthermore, the centromeric satellites identified by Avila Robledillo et al. were also mapped to specific chromosomes using FISH, and their reported positions are consistent with their centromeric locations in our assembled chromosomes. To provide comparison, we have updated Supplementary Table 11 to include this information. To assess assembly completeness at centromeres, we provide real gaps in centromere regions in chr2

and chr3, which are integrated into a single super-scaffold generated using Bionano data, confirming the correct sequence order (Supplementary Table 10).

4) What is the relationship between the open chromatin regions with genome assembly or population analysis? What is the logic for putting the open chromatin analysis in the manuscript.

Answer: Open chromatin analysis bridges genome assembly and population studies by validating assembly completeness (e.g., confirming regulatory regions like promoters) and annotating functional non-coding elements. In population analyses, it pinpoints regulatory variants (e.g., GWAS hits in open chromatin) linked to adaptive evolution or phenotypic diversity. Integrating this data strengthens the manuscript by connecting structural genomics (assembly) with functional and evolutionary mechanisms, offering mechanistic insights into how genetic variation drives traits—a critical layer often missing in standalone assembly or population analyses.

5) The author used a natural population with 406 accessions to conduct population analysis and highlighted this population as Fig. 2. Moreover, a selective sweeps analysis was performed and identified the VC1 locus is associated with a selection sweep during spring faba bean breeding. While to conduct GWAS analysis, the authors used a totally different population “ProFaba panel”, without describing it population, where at least the LD decay is essential for GWAS analysis.

Answer: We generated over 10-fold resequencing data for both spring and winter populations, which is adequate for population genomics analyses such as selective sweep scans. This analysis allowed us to identify genomic loci under selection specific to each population, e.g. *VC1* locus in spring lines. However, these populations are not ideal for mapping winter hardiness via GWAS. Therefore, we used the “ProFaba panel”, a diverse collection with rapid LD decay (672.9 Kbp) reported previously by Skovbjerg et al (2023). Further this panel contains about 13% (28 out 208) of high winter hardy genotypes providing sufficient power to detect significant marker-trait association through GWAS, which is limited in the detection of rare alleles or alleles with less 5% minor allele frequency. We have now described the population by clearly referencing Skovbjerg et al (2023).

6) Natural selection is also conducted for the “ProFaba panel”, where a Fst also identified two loci might under selection.

Answer: We performed Fst analysis using spring and winter population sequenced over 10-fold, which was mentioned in line number 234-240. The ProFaba panel was used only for GWAS analysis. We did not conduct selective sweep analysis using the ProFaba panel.

7) The d and e in Figure 3 (presumably) aim to indicate phenotypic differences between different types, but they lack statistical significance testing and annotations. The authors should review and add statistical tests for all similar instances in the manuscript.

Answer: Thanks for suggestion. We have added the Wilcoxon test for the significance testing between different phenotype in figure 3 d,e and figure 4 d,e,f.

8) In the analysis of cold tolerance, only 28 winter faba bean varieties were used, compared to 180 spring faba bean varieties. This may not fully represent the genetic diversity of winter varieties.

Answer: Faba bean exhibits a quantitative response to winter survival (Link W et al., 2010). GWAS has limited power in identifying rare alleles. But, as we answered previously, our panel includes over 10% (28 out 208) of winter-hardy genotypes, providing sufficient representation of winter growth habit related alleles. As a result, we were able to identify significant associations (Figure 3a). To find other important alleles contributing to winter hardiness, we used a panel consisting exclusively of winter types to conduct association mapping (Fig. 4a). We have used two different mapping populations - ProFaba to identify the main genetic differences between winter and spring types, and the Göttingen winter bean panel to identify further genetic opportunities to improving freezing tolerance within winter germplasm. That said, there could be additional sources of cold tolerance, like the Korean material, which could be further explored in future efforts.

9) GWAS analysis identified DREB1 as the candidate genes, and the authors further checked the expression profile of this gene with low temperature treatment. This gene has been reported in diverse plant species, what is the novel discovery here?

Answer: Briefly, the main novelty is that a single locus distinguishes winter and spring types in faba bean, as indicated in the title of the manuscript. The effect is dramatic with the single locus explaining most of the phenotypic variation for both late frost and winter survival rate as illustrated in Figure 3 d-e. While DREB1 has indeed been reported in model plants such as Arabidopsis and cereal grasses (e.g. wheat and barley) as a key regulator of cold stress responses, our study provides novel insights in the context of faba bean, a species that doesn't require strict vernalization like other winter crop (e.g. wheat). Our study represents a significant step toward understanding the genetic basis of cold tolerance in a legume crop where such mechanisms remain poorly characterized. Specifically, our GWAS pinpointed a major QTL for winter survival that colocalizes with DREB1, and we further validated its cold-inducible expression in winter-hardy genotypes. This combined evidence linking natural variation, trait association, and gene expression, establishes a specific DREB1 containing haplotype as a key candidate underlying adaptive variation in winter hardiness in faba bean, which has not been previously reported. Most importantly, our work demonstrates that natural allelic variation at DREB1 is associated with phenotypic differences in winter survival, which has direct application in breeding programs. The authors, O. Sass and G. Welna (faba bean breeders), have leveraged our results by developing markers to be used in their winter faba bean breeding program.

10) It is unclear whether the two materials used for the frost transcriptome sampling are part of the population. Do these materials' genotypes correspond to the spring and winter haplotype classification within the population? Why were materials not selected directly from the population for transcriptome sequencing? Using only two materials' transcriptome data to represent the entire population is very one-sided.

Answer: We appreciate the reviewer's concern. To strengthen our conclusion, we now conducted qRT-PCR analysis using five cold tolerant and susceptible faba bean genotypes and observed consistent results (Supplementary figure 13 and Supplementary table 24). Cold-tolerant genotypes were strategically selected based on prior field trials in Europe and South Korea. Notably, Korean genotypes, which were originally sourced from the USDA-National Plant Germplasm System (NPGS), survived winter temperatures as low as -17°C (Lyu et al., 2021), a level of hardiness rarely observed in Central and Northern Europe. We believed that these highly winter-hardy lines provided greater resolution for expression comparisons, which was obvious in our results. Additionally, the identification of the same genomic locus across diverse faba bean accessions further supports the robustness of our findings.

11) The subsequent analysis of the cold tolerance GWAS results only speculates and describes a small number of candidate genes using transcriptome data, which is clearly insufficient. The authors should include experimental validation and additional relevant analyses to further clarify the impact of these genes on cold tolerance and winter survival.

Answer: The lack of efficient transformation system in faba bean hinders to perform functional validation through gene editing system. However, we used robust genetic approach with multiple mapping populations to chart the genomic regions contributing to cold hardiness with greater resolution. Additionally, we validated our target genes using transcriptome assay with five cold tolerant individual accessions as suggested by reviewer 1. Our results and approach provided solid support for our conclusions within the scope of faba bean population genomics.

12) Line 293, what is the data used for expression analysis? Please clarify.

Answer: We have added the sample data we used in line 313.

13) Line 304, what is M1 and what is M2? Sound like the M2 model has more independent variables than M1, there is no wondering M2 improves prediction. Likelihood ratio testes could be performed to compare M1 and M2.

Answer: M1 and M2 are genomic best linear unbiased prediction (gBLUP) models that differ in whether the two loci identified by GWAS are included as fixed effects. We slightly revised the sentence for clarity. The likelihood ratio test (LRT) would be suited to compare M1 and M2 within the same data set, which would essentially reiterate the findings of the initial GWAS. However, across datasets, cross-validation is better suited to demonstrate that the mapped loci are not merely false positives but rather contribute meaningful genetic signal – as reflected by the improved predictive ability of the more complex model (M2).

14) Line 307, the heritability and genetic structure are different for different traits. The strength of the correlations between prediction and observation for loss of leaf etc, could not tell the prediction performance for freezing tolerance.

Answer: Discoloration and loss of turgidity are freezing symptoms that are frequently used to differentiate freezing tolerance among genotypes. While these are just two of several other traits related to freezing tolerance, each of which possibly exhibits different genetic architecture, they can distinguish well between levels of tolerance to freezing temperatures (Sallan et al., 2015; Arbaoui and Link., 2008). Therefore, selection based on the identified QTLs can help shift the population mean toward reduced freezing damage in winter-type faba bean. We have refined the final sentence for clarity.

15) Line 333, the syntenic conservation should be present in the result part.

Answer: Thanks for pointing this out. We have now moved the syntenic conservation to the results.

Reviewers' Comments:

Reviewer #1 (Remarks to the Author):

The authors clarified a little bit this part. Most of the conclusions are robust and valid with the data presented. Nevertheless, it is still not clear if the unbalance spring vs winter type (180/28) may be introducing some biases in the analysis analysis. The authors answered to the reviewer 2 "our panel includes over 10% (28 out 208) of winter-hardy genotypes, providing sufficient representation of winter growth habit related alleles. As a result, we were able to identify significant associations (Figure 3a)". Due the low diversity of the winter-hardy genotypes, it makes sense that it is going to produce a significant peak for the GWAS analysis (Figure 2a). Still the authors used a different population (Göttingen Winter Bean Population) to support their findings, but I found this part somewhat obscured in the manuscript. The qRT-PCR of other accessions is a good extra support, but again, somewhat is obscured and it does not appears clearly in the main manuscript.

Response: We appreciate the reviewer's constructive feedback and support of our research outcomes. Our reply that "our panel includes over 10% (28 out 208) of winter-hardy genotypes, providing sufficient representation of winter growth habit related alleles. As a result, we were able to identify significant associations (Figure 3a)" directly addresses the question of minor allele frequency for winter survival associated alleles. This is high enough to allow us to detect a reliable GWAS signal. In GWAS analyses, imbalances between groups with contrasting phenotypes for a given trait are common, as observed here for winter survival, and do not generally compromise the validity of results. As we mention in lines 273-276, the GWAS results are further supported by co-localising with the Fst signals differentiating spring and winter types. The winter and spring panels compared in the Fst analysis have very similar numbers of accessions (209 vs. 197), which might not have been clear in the previous version. We have clarified this point in line 276 of the revised main text.

Additionally, we conducted a separate GWAS analysis using only the winter panel for to identify opportunities for improvement within winter germplasm, but realise now that this could have been more clearly stated. We have revised the main text accordingly in line no. 311-312.

We have integrated the qRT-PCR results in the main manuscript by adding them as a new panel in figure 3 and description in line no. 297-299.

The authors have resolved all the points risen in this section, although an extra round of correction is recommended. There are some minor errors that should be fixed. For example line 485: "The BREAKER Long reads" should be BRAKER.

Response: Thanks for pointing this out. We have modified now.

Reviewer #2 (Remarks to the Author):

The authors had addressed my concerns and I have no further comments.

Response: We thank the reviewer.